# Dough Rheological Properties and Characteristics of Wheat Bread with the Addition of Lyophilized Kale (*Brassica oleracea* L. var. *sabellica*) Powder

Anna Korus [1], Mariusz Witczak [2], Jarosław Korus [3] and Lesław Juszczak [4,*]

1 Department of Plant Products Technology and Nutrition Hygiene, Faculty of Food Technology, University of Agriculture in Krakow, 30-239 Krakow, Poland
2 Department of Engineering and Machinery for Food Industry, Faculty of Food Technology, University of Agriculture in Krakow, 30-239 Krakow, Poland
3 Department of Carbohydrate Technology and Cereal Processing, Faculty of Food Technology, University of Agriculture in Krakow, 30-239 Krakow, Poland
4 Department of Food Analysis and Evaluation of Food Quality, Faculty of Food Technology, University of Agriculture in Krakow, 30-239 Krakow, Poland
* Correspondence: rrjuszcz@cyf-kr.edu.pl

**Abstract:** In this study, the effect of replacing 5 or 10% of wheat flour with lyophilized kale (*Brassica oleracea* L. var. *sabellica*) on the rheology of dough and bread characteristics (physical and textural properties, sensory acceptability, staling tendency) was evaluated. The farinographic analysis showed an increase in the development time, index of tolerance to mixing, and water absorption. The share of lyophilized kale in the dough affected changes in its rheological properties, e.g., increased the values of storage and loss moduli with a decrease in the value of the phase shift angle (tan δ) from 0.36 to 0.31 at 1 rad/s. A significant decrease in the values of instantaneous and viscoelastic compliance was also observed, and an increase in the value of zero shear viscosity. The incorporation of lyophilized kale into the dough caused a noticeable decrease in bread volume by about 10%, and porosity, by about 8%, despite the lack of statistical significance. Statistically significant changes were found in pore size and the presence of large pores > 5 mm² in the crumb, while pores density increased. The enrichment of bread with lyophilized kale influenced a decrease in the brightness of the crumb from 73.7 to 49.5 while increasing the proportion of yellow and green color as a result of a considerable increase in the content of chlorophyll pigments and carotenoids. Bread enriched with lyophilized kale had lower acceptability than the control bread. The enrichment of the bread with powdered kale also caused changes in the texture of the crumb, e.g., the hardness on the first day of the study was 2.14 N in the control bread, while in the bread with 10% kale content it was 6.46 N. In addition, the enriched bread showed a decrease in springiness, cohesiveness, and resilience.

**Keywords:** wheat bread; kale powder; dough rheology; bread quality; texture



## 1. Introduction

Bread is widely consumed around the world, therefore, plays a significant role in human nutrition [1]. Due to the growing awareness of health problems, the baking industry is trying to provide functional food with pro-health properties [2]. Studies on the nutritional value of bread are aimed at increasing the content of macronutrients (carbohydrates, proteins, fats, and dietary fiber), micronutrients (minerals and vitamins), and health-promoting components [2,3]. Attention is also paid to the sensory acceptability and shelf life of enriched products [4].

An example of the enrichment of baked goods with health-promoting ingredients can be the use of plant proteins, particularly proteins from legumes. Hoehnel et al. [5] developed a high-protein bread recipe in which wheat flour was partially replaced with ingredients

derived from broad beans and carob. Ayele et al. [4], on the other hand, used the addition of soy flour and cassava to wheat flour, and Viswanathan and Ho [6] used the addition of sprouted red kidney beans. Graça et al. [7] obtained good sensory acceptance for wheat bread enriched with yogurt and cottage cheese.

The addition of kale (*Brassica oleracea* L. var. *acephala*) to bread may be an interesting alternative for the development of innovative bakery foods. Kale is a biennial vegetable belonging to the *Brassicaceae* family, which has been cultivated for centuries in many countries. It is consumed especially in the US, Japan, and the Mediterranean region [8]. Kale has been extensively studied for its nutritional value [9–11] attributed to the high content of bioactive components, including phenolic compounds, glucosinolates, chlorophylls, and carotenoids [12,13]. Kale leaves also contain quite a lot of minerals (Ca, K, Fe, Mg), vitamins (C, E), and unsaturated fatty acids [14,15]. It should also be noted that kale has a relatively high protein content. Korus [16] showed that it is a high-value protein containing significant amounts of glutamic acid, aspartic acid, proline, and leucine. Kale is also a very good source of dietary fiber, the consumption of which reduces the risk of cancer and cardiovascular diseases [17–19].

The consumption of kale has been shown to have antioxidant, anti-carcinogenic, and protective effects on the cardiovascular system and gastrointestinal tract [20–22]. Kale has been found to protect against high-fat diet (HFD)-induced dysfunction through mechanisms including lipid metabolism, endotoxemia, and inflammation [23]. Due to its health-promoting properties, it is used, among others, for the production of frozen foods, chips, drinks (smoothies), and vegetable pastes, but also in the production of dietary supplements or in the cosmetics industry. Thus, the use of this raw material in Thus, the use of this vegetable in bread baking can fit into the current nutritional trend related to the enrichment of products with ingredients with high health-promoting properties. For example, Klopsch et al. [24] enriched wheat bread in kale microgreens and fresh young kale leaves.

Therefore, the aim of our study was to investigate the effect of the addition of fully ripe kale leaves, in the form of lyophilizate, on the wheat dough, the quality, and the shelf-life of the resulting bread.

## 2. Materials and Methods

### 2.1. Materials

The materials were wheat flour (ash content max. 0.58%) (PZZ Krakow, Poland), salt, lyophilized yeast (Lesaffre, France), and water. Powdered lyophilized kale leaves with the following composition were used as an enrichment ingredient: protein $28.9 \pm 0.92$ g/100 g, dietary fiber $23.56 \pm 0.21$ g/100 g, including soluble fraction $5.27 \pm 0.08$ g/100 g, insoluble fraction $18.29 \pm 0.12$ g/100 g, starch $12.07 \pm 0.25$ g/100 g, total sugars $11.39 \pm 0.39$ g/100 g, ash $7.50 \pm 0.11$ g/100 g, total polyphenols $1236 \pm 16$ mg/100 g, chlorophylls (a + b) $766.7 \pm 12.7$ mg/100 g, carotenoids $205.0 \pm 7.4$ mg/100 g [25–27]. Kale leaves purchased from the local supermarket (Kraków, Poland) was blanched in water at 98 °C for 2.5 min. They were then frozen on freeze-dryer trays in an air-vented chamber (Feutron GmbH, Langenwetzendorf, Germany). Freezing was carried out at −40 °C for 90 min. The lyophilization process was started at a temperature of −25 °C (CHRIST Gamma 1–16 LSC freeze-dryer, Shrewsbury, UK). A moisture content of about 5% was achieved in the dried material after 30 h using. After drying, the material was ground in a GM200 grinder (Retsch, Haan, Germany), sieved, and stored in hermetically sealed vessels.

### 2.2. Dough and Bread Making

The dough was obtained from 350 g of wheat flour, 10.5 g of yeast, 7 g of salt, and 215 g of water. In enriched bread, lyophilized kale was added instead of 5 or 10% wheat flour. The amount of water added was optimized by the back extrusion method using a TA-XT2plus texturometer (Stable Micro Systems, Godalming, UK).

The dough was mixed for 10 min in spiral mixer type SP 12 (Diosna Dierks & Söhne GmbH, Osnabrück, Germany) and then fermented at 35 °C for 30 min. After fermentation,

portions of 50 g dough were placed in molds and subjected to final fermentation for 30 min. The bread was baked in a at 230 °C for 20 min. After baking, the loaves were removed from the molds and cooled. Bread for storage tests was kept in polyethylene bags at room temperature.

### 2.3. Dough Rheological Tests

Farinographic evaluation of dough rheology was performed using a Farinograph-E (Brabender, Duisburg, Germany) equipped with a 50 g flour mixer, according to AACC standard 54-21.02 [28].

The rheological tests were carried out using MARS II oscillating rheometer (ThermoFisher Scientific, Waltham, MA, USA) at a temperature of 25 °C equipped with parallel serrated plates with a diameter of 35 mm and a gap size of 1 mm. The investigated samples were placed in the rheometer for 5 min for stress relaxation and temperature stabilization.

The spectra were recorded in the range of linear viscoelasticity at a constant strain amplitude of 0.3% in the angular frequency of 1–100 rad/s. The power law equations were used to describe the data obtained [29]:

$$G'(\omega) = K' \cdot \omega^{n'} \tag{1}$$

$$G''(\omega) = K'' \cdot \omega^{n''} \tag{2}$$

$G'$—storage modulus (Pa), $G''$—loss modulus (Pa), $\omega$—angular frequency (rad/s), $K'$, $K''$, $n'$, $n''$—constants parameters.

Both tests, creep and recovery were carried out with constant creep strain $\tau_0 = 5$ Pa for 150 s. The recovery test was continued for 600 s. The Burgers model was used to describe the data [30]:

$$J(t) = J_0 + \frac{t}{\eta_0} + J_1 \cdot \left(1 - \exp^{-\frac{t}{\lambda_{ret}}}\right) \text{for creep phase} \tag{3}$$

$$J(t) = \frac{t}{\eta_0} - J_1 \cdot \left(1 - \exp^{-\frac{t_1}{\lambda_{ret}}}\right) \cdot \exp^{-\frac{t}{\lambda_{ret}}} \text{ for recovery phase} \tag{4}$$

$J$—compliance ($\text{Pa}^{-1}$), $J_0$—instantaneous compliance ($\text{Pa}^{-1}$), $J_1$—viscoelastic compliance ($\text{Pa}^{-1}$), $\eta_0$—zero shear viscosity (Pa·s), $\lambda_{ret}$—retardation time (s), and $t_1$—time after which the stress was removed (s).

### 2.4. Evaluation of Composition and Physical Properties of Bread

Protein, dietary fiber, as well as total phenolic content were determined in the bread samples [25–27].

The bread volume was determined with a Volscan Profiler 600 (Stable Micro Systems, Godalming, UK). Investigation of crumb structure characteristics: slices of bread 2 cm thick were scanned (Plustek S-12, Plustek, Taiwan), and the results were obtained using the ImageJ program (NTH, Barron, WI, USA). The average pore size, pore density, porosity, and number of pores larger than 5 $\text{mm}^2$ were determined [29].

### 2.5. Bread Crumb Color and the Content of Pigments

CIE L*a*b* crumb color analysis was performed according to the reflectance method with a Konica MINOLTA CM-3500d spectrometer (Konica Minolta Inc., Tokyo, Japan) (illuminant D65, observer 10°, geometry d/8).

### 2.6. Sensory Test of Bread

The test of sensory attributes of bread was performed in accordance with the Polish Standard PN-A-74108:1996 [31]. The 15 semi-trained panelists evaluated the following parameters: external appearance (max 5 points), crust color (max 3 points), thickness (max 4 points), other crust attributes (max 4 points), crumb-elasticity (max 4 points), porosity (max 3 points), other crumb attributes (max 3 points), and taste and aroma (max 6 points).

### 2.7. Texture Test of Bread

Texturograms characterizing the properties of the breadcrumb were obtained using a TA-XT2plus texturometer (Stable Micro Systems, Godalming, UK). Crumb samples in the shape of a cylinder (1 cm in diameter, 2 cm in height), taken from the central part of the loaves, were compressed twice with an aluminum P/35 cylindrical probe until 50% deformation at a compression speed of 5 mm/s. From the obtained texturograms, the crumb hardness, chewiness, springiness, cohesiveness, and resilience were determined. Calculations were carried out using Texture Exponent software (Stable Micro Systems, Godalming, UK). The analyses were conducted over three consecutive days.

### 2.8. Thermal Evaluation of Bread

The thermal characteristics of the crumb were performed using the DSC 204F1 Phoenix differential scanning calorimeter (Netzsch GmbH, Selb, Germany). In or determine the freezable water in bread after baking [32], about 12 mg of the crumb was sealed in hermetic aluminum vessels and frozen to $-60\ ^{\circ}$C at a speed of 10 $^{\circ}$C/min. The frozen samples were held for 5 min at this temperature, followed by heating to 50 $^{\circ}$C at speed indicated above. The standard was the empty aluminum vessel. Temperatures: the beginning of transformation $T_O$, maximum of transformation $T_P$, end of transformation $T_E$, as well as enthalpy of melting, were determined using Proteus Analysis software (Netzsch GmbH, Selb, Germany). The content of freezable water (FW) expressed in g/g was calculated using the following formula:

$$FW = \frac{\Delta H}{\Delta H_0} \tag{5}$$

$\Delta H$—enthalpy of ice melting (J·g$^{-1}$), $\Delta H_0$—latent heat of ice melting (334 J·g$^{-1}$).

In order to determine the staling rate of the bread crumb during the three-day storage tests, crumb samples (approximately 12 mg) were sealed in hermetic aluminum vessels and heated in a calorimeter to 100 $^{\circ}$C at a speed of 10 $^{\circ}$C/min. The standard was the empty aluminum vessel. The values of temperature and enthalpy of the thermal transformation were obtained from the Proteus Analysis program (Netzsch GmbH, Selb, Germany). The enthalpy values are given in J/g of dry weight.

### 2.9. Statistical Analysis

Data were analyzed using a one-way analysis of variance and Duncan's post hoc test at the significance level of 0.05. In the case of storage tests, a two-way analysis of variance was used for the texture analysis and thermal properties. Statistica v. 13.0 (StatSoft Inc., Tulsa, OK, USA) was used to analyze the data.

## 3. Results and Discussion

### 3.1. Dough Rheological Tests

Evaluation of the effect of various recipes or technological modifications on the wheat dough is most often conducted by farinographic analysis. The kale powder increased the dough development time significantly ($p < 0.05$), but only for the higher $-10\%$ addition (Figure 1).

In contrast, recipe modification had no significant ($p < 0.05$) effect on dough stability time values. The addition of lyophilized kale to the dough resulted in a significant ($p < 0.05$) increase in the mixing tolerance index and water absorption. The increase in the value of the mixing tolerance index indicates a gluten weakening, which may be due to the introduction of compounds with different molecular weights and water-binding capacity into the dough, interfering with the formation of a gluten network. This confirms previous observations about the weakening of the dough structure due to the addition of various powdered vegetable raw materials. The dough structure weakening may be due to the dilution of structural proteins on the effect of replacing part of the flour with a gluten-free vegetable component [33], as well as mutual competition between gluten proteins and high-molecular-weight fiber components introduced with powdered kale for available water. Wang et al. [34] found an increase in water absorption in the dough and

a weakening of the gluten network structure when celery powder was introduced into the recipe. Bigne et al. [35] noted that replacing wheat flour with mesquite flour causes changes in water absorption, and the resulting dough has less stability compared to wheat dough, which is affected by the weak development of the gluten network.

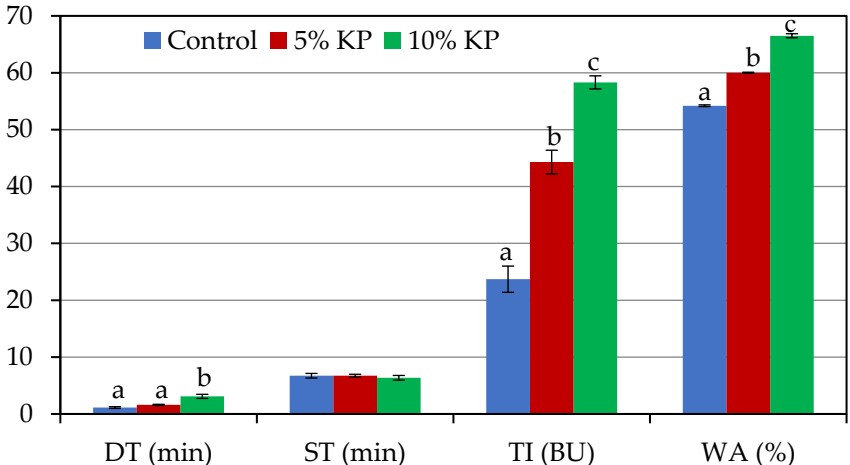

**Figure 1.** Results of faringographic analysis: DT—development time, ST—stability time, TI—tolerance index, WA—water absorption; means marked with the same letters do not differ statistically at 0.05 level of confidence.

Also, small deformation rheological methods are used to evaluate various types of doughs and the impact of a recipe or technological changes. The obtained functional relationships $G'$, $G'' = f(\omega)$ indicate that the values of the moduli of $G'$ are higher than those of $G''$ ($G' > G''$) over the entire range of angular frequency. This indicates the domination of elastic characteristics over viscous ones, which is also supported by the tangent $\delta$ values ranging from 0.31 to 0.43 (Figure 2b). The addition of lyophilized kale resulted in an increase in the values of both moduli (Figure 2a) with a decrease in the tan $\delta$ values (Figure 2b). These observations confirmed the parameters of the power equations describing the sweep frequency curves (Table 1).

**Table 1.** Rheological characteristics of the tested wheat and wheat-kale composite (KP) doughs.

| Parameter | Control | 5% KP | 10% KP | Anova-*p* |
|---|---|---|---|---|
| $K'$ (Pa·s$^{n'}$) | 6194 ± 264.6 [a] | 12,535 ± 216.1 [b] | 19,285 ± 659.0 [c] | <0.001 |
| $n'$ | 0.211 ± 0.003 [c] | 0.205 ± 0.002 [b] | 0.190 ± 0.002 [a] | <0.001 |
| $r^2$ > | 0.999 | 0.994 | 0.996 | |
| $K''$ (Pa·s$^{n''}$) | 2306 ± 59.9 [a] | 4369 ± 103.4 [b] | 6294 ± 145.1 [c] | <0.001 |
| $n''$ | 0.222 ± 0.007 [c] | 0.205 ± 0.002 [b] | 0.193 ± 0.006 [a] | 0.002 |
| $r^2$ > | 0.973 | 0.958 | 0.976 | |
| tan $\delta$ (at 1 Hz) | 0.364 ± 0.003 [c] | 0.330 ± 0.001 [b] | 0.312 ± 0.010 [a] | <0.001 |
| $J_0 \times 10^4$ (Pa$^{-1}$) | 3.38 ± 0.10 [c] | 1.55 ± 0.13 [b] | 0.93 ± 0.09 [a] | <0.001 |
| $J_1 \times 10^4$ (Pa$^{-1}$) | 6.27 ± 1.17 [c] | 2.59 ± 0.48 [b] | 1.28 ± 0.15 [a] | <0.001 |
| $\eta_0 \times 10^{-6}$ (Pa·s) | 0.58 ± 0.10 [a] | 1.39 ± 0.24 [b] | 1.65 ± 0.37 [b] | 0.006 |
| $\lambda$ (s) | 128 ± 25 [a] | 116 ± 19 [a] | 77 ± 20 [a] | 0.062 |
| $r^2$ > | 0.987 | 0.987 | 0.990 | |

Mean value of three replications ± standard deviation. Means in rows marked with the same letters do not differ statistically at a 0.05 level of confidence.

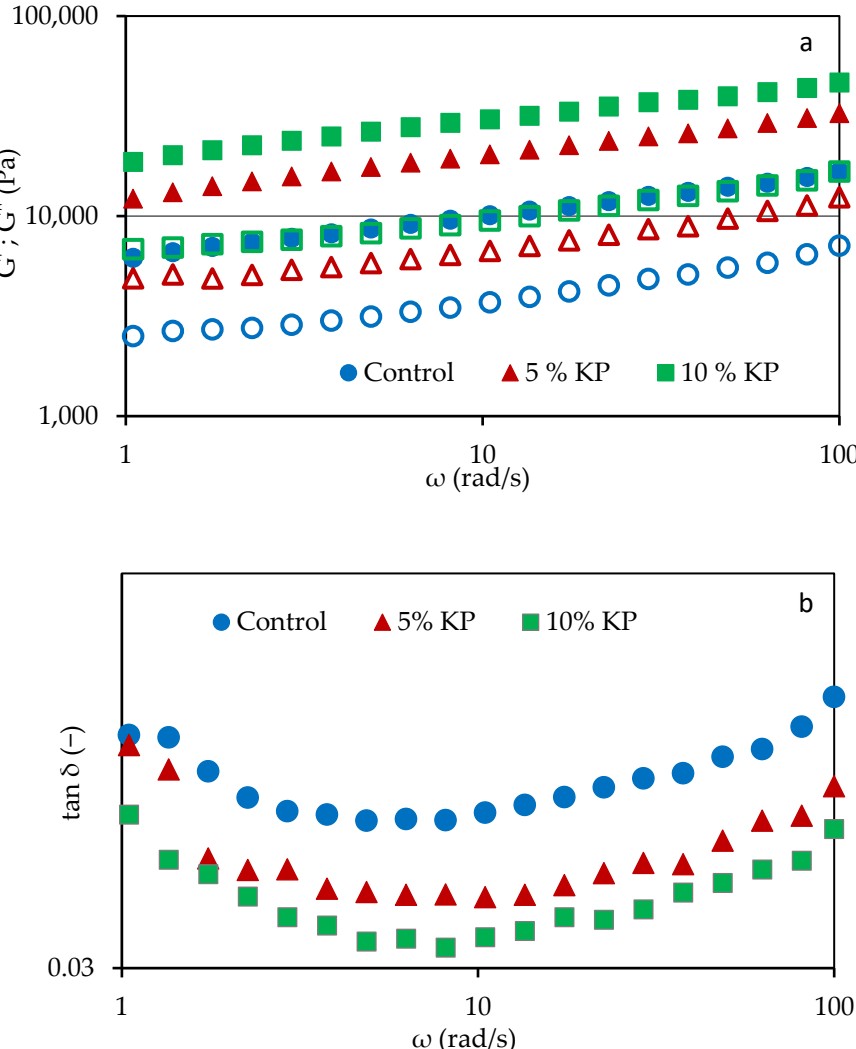

**Figure 2.** Mechanical spectra (**a**) (G′—filled symbols, G″—empty symbols) and shift angle tangent (tan δ = G″/G′) (**b**) of the investigated doughs.

The values of the K′ parameter, related to the beginning value of the module G′, significantly ($p < 0.05$) increased with an increase in the amount of kale lyophilizate, while the values of the n′ parameter significantly ($p < 0.05$) decreased with the increase in the share of kale. This indicates a decrease in the reliance of the G′ modulus from angular frequency, thus the dough structure strengthening in the range of small deformations. Similarly, for the parameter K″ for which a significant ($p < 0.05$) increase was observed, and the parameter n″ for which a significant decrease ($p < 0.05$) in value was found due to the addition of lyophilized kale. Since the increase in the values of both moduli was not proportional, these changes also exerted an effect on the values of tan δ, which significantly ($p < 0.05$) decreased (Figure 2b, Table 1), which indicates a strengthening of the dough.

The curves in the creep phase reflect the changes in susceptibility over time at a constant stress value, while the curves of the recovery phase at zero strain reflect the energy stored in the material structure that is recovered (Figure 3). The addition of lyophilized kale to the dough resulted in a decrease (significant at $p < 0.05$) in susceptibility to stress the greater, the more flour was replaced with powdered kale. This is reflected in the values of instantaneous ($J_0$) and viscoelastic compliance ($J_1$), which significantly ($p < 0.05$) decreased as the proportion of lyophilized kale in the recipe increased. The strengthening of the dough in the range of small deformations is also confirmed by zero shear viscosity ($\eta_0$), which significantly ($p < 0.05$) increased with an increasing amount of lyophilized kale (Table 1). In contrast, there was no significant impact of the kale powder on retardation time (Table 1).

According to Mastromatteo et al. [36], the presence of non-hydrated vegetable flour in wheat dough resulted in competition for water by protein and starch forming the dough structure, which could lead to a tougher dough. The results obtained clearly indicate the dependence of the observed properties on the magnitude of the applied stress. In the range of small deformations, a strengthening of the dough structure was observed, resulting from the introduction into the system of high-molecular-weight components with high swelling capacity, which, despite the deterioration of the gluten network, maintain the structure of the dough analogous to that in rye bread. On the other hand, in the range of large deformations, due to the weakened gluten structure resulting from its smaller amount and limited hydration, the system is more susceptible to applied stress.

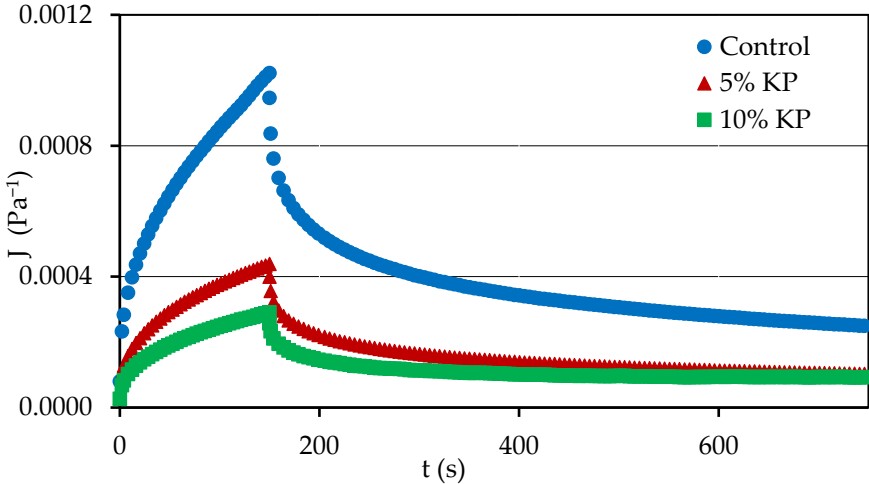

**Figure 3.** Creep and recovery curves of control dough and samples with kale powder at different concentrations.

### 3.2. Composition and Physical Properties of Bread

Bread enrichment is aimed at improving its nutritional properties and supplementing the diet with bioactive compounds. However, changes made to the formulation should not adversely affect the other properties of the bread, including its sensory acceptability. One method of bread enrichment is the introduction of vegetable raw materials into the recipe, which enrich the bread mainly with dietary fiber and bioactive compounds [34,36–38]. The protein content in the control bread was 9.76 g/100 g, which confirms previous observations regarding various types of bread [3]. The highest 10% share of lyophilized kale resulted in a non-significant ($p > 0.05$) increase in protein content to 10.12 g/100 g. The lack of significant effect on the content of this component is due to the fact that wheat flour itself contains a significant amount of protein. The content of dietary fiber in the control bread was 2.62 g/100 g, including 0.93 g/100 g of the soluble fraction and 1.69 g/100 g of the insoluble fraction. The data obtained are slightly lower than the values reported for various types of bread by Fraś et al. [3]. The addition of kale lyophilizate resulted in a significant ($p < 0.05$) increase in the content of dietary fiber and its fractions. In the bread containing 10% of kale, 3.77 g of fiber per 100 g was found, including 1.14 g/100 g of the soluble fraction and 2.63 g/100 g of the insoluble fraction. The control bread analyzed had a relatively low content of total polyphenols at 7.16 mg/100 g due to their low content in wheat flour [39]. The addition of 5% and 10% lyophilized kale resulted in a significant ($p < 0.05$) increase in total polyphenol content to 48.87 mg/100 g and 87.44 mg/100 g, respectively. This confirms the previous observations concerning the possibility of enriching bread with polyphenols through the addition of raw vegetable materials.

The essential parameter characterizing the bread and providing an opportunity to assess the influence of the recipe composition is the volume and specific volume of the loaves.

The data summarized in Table 2 indicate that the lyophilized kale addition caused a small and non-significant ($p > 0.05$) decrease in the volume and specific volume of the bread. The available literature data indicate that replacing wheat flour with other ingredients, including lyophilized vegetables, can result in a decrease in loaf volume [37]. In turn, Wang et al. [34], who enriched bread with celery powder, obtained similar results. Dhillon et al. [40] found a decrease in bread volume when vegetable pastes were introduced into the recipe. Also, Lafarga et al. [38] observed a decrease in the specific weight of bread enriched with a broccoli-based preparation. In a study by Bigne et al. [35] and González-Montemayor et al. [41], the addition of mesquite flour to wheat dough resulted in a decrease in bread volume because the presence of mesquite interfered with the formation of the bread structure by weakening the gluten network. Ptitchkina et al. [42] enriched bread with pumpkin powder, observing an increase in volume at lower doses of this ingredient and a decrease in volume at higher doses. The addition of lyophilized kale to the dough resulted in some changes in the crumb structure, as reflected in the values of parameters determined by digital image analysis (Table 2). The presence of lyophilized kale caused a significant ($p = 0.008$) reduction in average pore diameter in comparison with the control bread, while the amount of kale was not significant. There was also a decrease in porosity, but the differences were non-significant ($p > 0.05$). In contrast, a significant ($p = 0.021$) increase in pore density with a simultaneous significant ($p = 0.004$) decrease in the number of large pores ($>5$ mm$^2$) was observed for bread with a higher addition of lyophilized kale. These results indicate that the crumb of bread enriched in this way is characterized by a greater number of fine pores, which may favorably influence the acceptance of its structure.

**Table 2.** Physical characteristics of the investigated wheat and wheat-kale (KP) composite bread.

| Parameter | Control | 5% KP | 10% KP | Anova-$p$ |
|---|---|---|---|---|
| Volume (cm$^3$) | $120 \pm 7$ [a] | $114 \pm 5$ [a] | $108 \pm 8$ [a] | 0.060 |
| Specific volume (cm$^3 \cdot$g$^{-1}$) | $2.86 \pm 0.18$ [a] | $2.70 \pm 0.11$ [a] | $2.55 \pm 0.23$ [a] | 0.054 |
| Average pore size (mm$^2$) | $0.973 \pm 0.055$ [b] | $0.890 \pm 0.026$ [a] | $0.810 \pm 0.036$ [a] | 0.008 |
| Porosity | $0.443 \pm 0.006$ | $0.420 \pm 0.010$ | $0.407 \pm 0.029$ | 0.113 |
| Cell density (cm$^{-2}$) | $44.9 \pm 3.1$ [a] | $49.9 \pm 0.9$ [ab] | $53.6 \pm 3.4$ [b] | 0.021 |
| Percentage of pores > 5 mm$^2$ | $3.68 \pm 0.73$ [b] | $2.75 \pm 0.20$ [b] | $1.48 \pm 0.29$ [a] | 0.004 |

Mean value of four replications ± standard deviation. Means in rows marked with the same letters do not differ statistically at a 0.05 level of confidence.

The determined color parameters (Figure 4a) also indicate significant changes in the appearance of the crumb. The highest value of the crumb L* parameter had the control bread indicating the brightness of the sample, and the presence of kale significantly ($p < 0.05$) reduced it. The a* parameter of the control sample showed positive values indicating the contribution of the red color. In contrast, the presence of lyophilized kale caused a shift in the value of this parameter toward negative values, indicating an increasing proportion of green color. Similarly, in the case of the parameter b* indicating the contribution of yellow color, bread with kale powder had significantly ($p < 0.05$) higher values for this parameter than the control sample, but without a clear effect of the enrichment level itself (Figure 4a).

The changes in crumb color characteristics are due to the introduction of kale powder, the natural plant tissue pigments. The contents of chlorophylls a, b, their sum, and carotenoids in the control bread and samples with lyophilized kale are shown in Figure 4b. As the amount of kale lyophilizate increased, the content of individual pigments increased significantly ($p < 0.05$). The obtained results on color change reflect the literature data, which indicate that the use of vegetable or vegetable industry waste as ingredients in bakery products significantly affects their color [43]. Krupa Kozak et al. [44] observed a significantly darker color of the crumb and crust of gluten-free bread enriched with a broccoli-based preparation. In addition, such bread showed significantly lower negative values of the a* parameter and significantly higher values of the b* parameter than control bread, indicating an increase in the proportion of green and yellow color. Also, according to

Lafarga et al. [38], bread enriched with a broccoli-based preparation has a higher intensity of green color, both in the crust and crumb. However, according to Drabińska et al. [45], this can adversely affect the acceptability of such products.

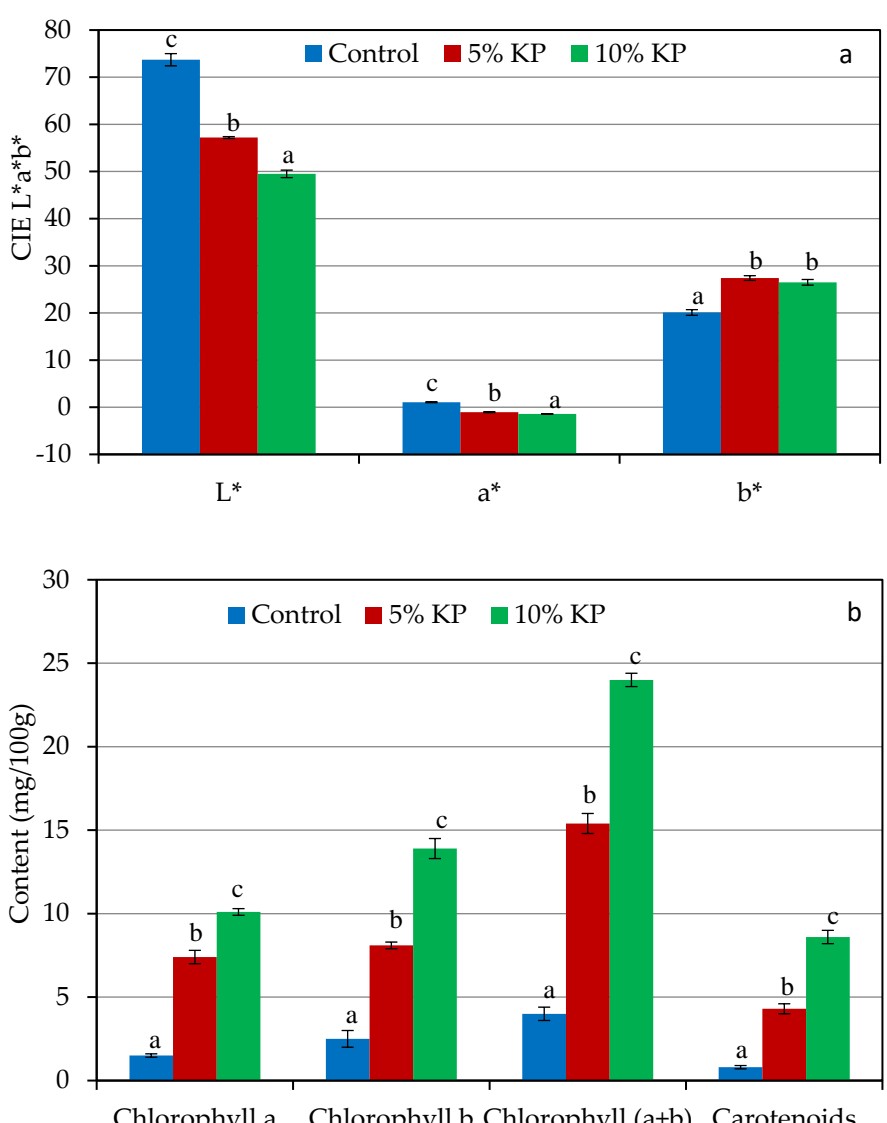

**Figure 4.** Color parameters (**a**) and chlorophyll and carotenoids content (**b**) of the investigated breadcrumbs; means marked with the same letters do not differ statistically at 0.05 level of confidence.

### 3.3. Bread Acceptability

The highest acceptability in terms of overall appearance was achieved by the control bread, and the modification of the recipe influenced a decrease in the acceptability of this attribute, but no significant ($p > 0.05$) differences were found.

In the evaluation of crust color, a downward trend in acceptability was noted, but also the differences were non-significant ($p > 0.05$). The crust of the enriched bread was characterized by a darker than control bread, greenish color, less characteristic of wheat bread, which reduced its acceptability. The presence of lyophilized kale also had a non-significant ($p > 0.05$) impact on the acceptability of crust thickness. In contrast, other crust characteristics were rated significantly ($p < 0.05$) lower than that of the control sample but with no variation in terms of lyophilized kale level. However, it can be assumed that the decisive influence on the results of this evaluation was the greenish color of the crust. A similar trend was found when evaluating the acceptability of crumb characteristics: springiness, porosity, and other properties. It was found, in all cases, a decreasing trend in

acceptability, but with non-significant ($p > 0.05$) differences. On the other hand, the decline in acceptability in the case of taste and aroma was significant (Table 3), but the large values of standard deviations for the mean values resulted in no significant differences between them. This indicates that the panelists did not agree in assessing the acceptability of aroma and taste, which may be due to individual differences in preferences. These results are reflected in the work of other authors. Wang et al. [34] observed a significant adverse effect of the share of celery powder on the acceptability, both overall and of individual sensory attributes. Similarly, Drabińska et al. [45] noted that the share of broccoli-based powder in gluten-free cookies results in a significant decrease in product acceptability due to its green color, increase in hardness, and intense smell and taste. Danza et al. [33] also observed a decrease in the acceptability of individual organoleptic characteristics of baked goods enriched with yellow bell pepper flour. Also, Czaja et al. [39] observed a decrease in the acceptability of the taste of bread enriched with onion extract. On the other hand, Hobbs et al. [46] indicate that enriching baked goods with vegetable-based raw materials is an appropriate strategy for promoting and increasing their consumption, but the sensory acceptability of such products is closely connected with the kind of vegetables used. Probably, the appropriate promotion and health information for this type of product could favorably influence its sensory acceptability.

**Table 3.** Sensory characteristics of the investigated wheat and wheat-kale (KP) composite bread.

| Property | | Control | 5% KP | 10% KP | ANOVA-$p$ |
|---|---|---|---|---|---|
| Appearance | | 4.9 ± 0.4 [a] | 4.1 ± 1.2 [a] | 3.9 ± 1.6 [a] | 0.075 |
| Crust | Color | 2.8 ± 0.4 [a] | 2.3 ± 0.8 [a] | 2.2 ± 1.0 [a] | 0.104 |
| | Thickness | 3.8 ± 0.4 [a] | 3.5 ± 0.5 [a] | 3.5 ± 0.5 [a] | 0.150 |
| | Others | 3.9 ± 0.3 [b] | 3.5 ± 0.5 [a] | 3.5 ± 0.5 [a] | 0.026 |
| Crumb | Elasticity | 3.9 ± 0.3 [a] | 3.7 ± 0.5 [a] | 3.4 ± 1.1 [a] | 0.108 |
| | Porosity | 2.9 ± 0.3 [a] | 2.7 ± 0.5 [a] | 2.6 ± 0.8 [a] | 0.279 |
| | Others | 2.8 ± 0.4 [a] | 2.3 ± 0.8 [a] | 2.3 ± 0.8 [a] | 0.127 |
| Smell and taste | | 5.3 ± 1.5 [a] | 3.5 ± 2.6 [a] | 3.6 ± 2.7 [a] | 0.082 |

Mean value of fifteen replications ± standard deviation. Means in rows marked with the same letters do not differ statistically at a 0.05 level of confidence.

### 3.4. Bread Texture

The texture of the bread is an essential criterion in assessing its quality. Analysis of the variation of its individual parameters allows for the assessment of the impact of a recipe or technological modifications and storage time on the quality. On the first day, the crumb of the control bread had the lowest hardness of 2.14 N (Table 4). The addition of lyophilized kale caused a significant ($p < 0.05$) increase in crumb hardness to a level of 6.46 N at a 10% bread enrichment level. In the case of resilience, springiness, and cohesiveness, lyophilized kale reduced these parameters the greater the level of enrichment, but only the highest 10% share of lyophilized kale caused statistically significant ($p < 0.05$) changes (Table 4). The springiness characteristic of fresh bread is important for consumers, as well as sufficient cohesiveness, to reduce the tendency for crumbling, and a decrease in the values of these parameters indicates a deterioration in the quality of the bread. Since chewiness is closely connected with hardness, the effect of lyophilized kale was analogous, and its values increased significantly ($p < 0.05$) as the amount of kale increased (Table 4). The results obtained are reflected in the literature data. Wang et al. [34] observed a significant decrease in springiness and resilience and an increase in chewiness and hardness of bread enriched with celery powder. Also, Ranawana et al. [47] found an increase in chewiness and hardness as well as a decrease in crumb cohesiveness of bread enriched with various lyophilized vegetables. Dhillon et al. [40] noted an increase in the hardness of the crumb of bread enriched with vegetable pastes.

During storage, bread undergoes complex structural changes related to water migration, starch retrogradation, and reorganization of protein-starch complexes. These changes are generally referred to as staling, and their effects result in a deterioration of the consumer

acceptability of bread. In instrumental measurement, bread staling is manifested by increasing hardness and chewiness and decreasing springiness and resilience [48]. The crumb of bread with 10% of lyophilized kale on the third day of the study had the highest hardness, and the share of kale, storage time, and the interaction of both factors had a significant ($p < 0.05$) impact on this parameter (Table 4). However, it is important to note the differences in the increment of crumb hardness values. For the control bread, the increase in hardness over three days of storage was 3.7 times, while for the crumb containing 10% lyophilized kale, it was 2.3 times. This indicates that the harder the crumb is after baking, the lower the growth rate of this parameter. An analogous trend was observed for chewiness (Table 4). On the other hand, the values of the other parameters characterizing texture decreased during storage, but the statistical variation was smaller, and for cohesiveness and resilience, the level of kale had a non-significant ($p > 0.05$) effect.

**Table 4.** Texture parameters of the investigated wheat and wheat-kale (KP) composite bread.

| Sample | Day | Hardness (N) | Resilience | Springiness | Cohesiveness | Chewiness (N) |
|---|---|---|---|---|---|---|
| Control | 1 | 2.14 ± 0.02 [a] | 0.50 ± 0.01 [f] | 1.00 ± 0.00 [d] | 0.91 ± 0.02 [e] | 2.03 ± 0.18 [a] |
| | 2 | 4.89 ± 0.62 [c] | 0.33 ± 0.01 [bcd] | 0.97 ± 0.01 [c] | 0.77 ± 0.05 [ab] | 3.60 ± 0.23 [c] |
| | 3 | 7.90 ± 1.47 [e] | 0.28 ± 0.02 [a] | 0.96 ± 0.01 [c] | 0.71 ± 0.04 [a] | 5.33 ± 0.83 [d] |
| 5% KP | 1 | 3.29 ± 0.09 [b] | 0.48 ± 0.00 [f] | 0.99 ± 0.01 [d] | 0.89 ± 0.02 [de] | 2.91 ± 0.07 [b] |
| | 2 | 6.02 ± 0.40 [d] | 0.35 ± 0.01 [d] | 0.96 ± 0.01 [c] | 0.80 ± 0.05 [bc] | 4.60 ± 0.23 [d] |
| | 3 | 9.60 ± 0.71 [f] | 0.30 ± 0.02 [ab] | 0.95 ± 0.00 [ab] | 0.73 ± 0.07 [ab] | 6.63 ± 0.10 [e] |
| 10% KP | 1 | 6.46 ± 0.36 [de] | 0.45 ± 0.01 [e] | 0.97 ± 0.02 [c] | 0.85 ± 0.04 [cd] | 5.36 ± 0.35 [d] |
| | 2 | 11.7 ± 0.96 [fg] | 0.34 ± 0.02 [cd] | 0.96 ± 0.00 [bc] | 0.75 ± 0.05 [ab] | 8.49 ± 1.30 [f] |
| | 3 | 14.9 ± 0.94 [g] | 0.31 ± 0.03 [abc] | 0.94 ± 0.01 [a] | 0.72 ± 0.06 [ab] | 10.0 ± 1.51 [f] |
| Two-way ANOVA-*p* | | | | | | |
| Factor A (level) | | <0.001 | 0.055 | 0.001 | 0.235 | <0.001 |
| Factor B (time) | | <0.001 | <0.001 | <0.001 | <0.001 | <0.001 |
| Factor A × Factor B | | <0.001 | <0.001 | 0.326 | 0.413 | <0.001 |

Mean value of three replications ± standard deviation. Means in columns marked with the same letters do not differ statistically at a 0.05 level of confidence.

*3.5. Thermal Properties of the Crumb*

A sufficient lowering of the ambient temperature causes water to freeze in the crumb structure. Re-heating induces the melting of ice crystals observed on thermograms as an endothermic transformation, the onset of which reaches about −20 °C [32]. The values of parameters characterizing the endothermic transformation related to the melting of frozen water crystals are summarized in Table 5. The values of the transformation onset temperature varied from −10.3 to −9.0 °C and were lower for the control, with only the level of enrichment with kale having a significant ($p < 0.05$) effect. Similar values, ranging from −8.5 to −8.1 °C, for the onset of transformation were obtained by Ribotta and Le Bailc [32]. A greater statistical diversity was observed in the case of maximum transformation temperature, the values of which generally increased with increasing kale level and decreased for individual samples with storage time, but for the highest proportion of lyophilized kale, the changes were non-significant ($p > 0.05$). A similar trend was found for end-of-transformation temperature. Both the amount of kale and storage time had a significant impact on this parameter (Table 5).

The values of transformation enthalpy increased with the level of addition of kale and decreased during the study, and the share of kale, storage time, and the interaction of both factors had a significant ($p < 0.05$) impact on the results. The observed changes in transformation enthalpy values correlate ($r = 0.99$) with the amount of freezable water, the content of which also increased with the amount of kale introduced and decreased during storage. These results correlate closely with the amount of water introduced into the dough to maintain its consistency. Since the enriching addition of lyophilized kale proved to be highly water-absorbent, the selection of the appropriate amount of water for each recipe was carried

out based on the results of faringraph analysis. On the other hand, the decrease in freezable water content during storage is presumably due to the loss of crumb moisture, which moves toward the crust, and the incorporation of water molecules into the crystalline structure of starch, which forms during the staling of the bread. Similar results on changes in freezable water content during bread storage were obtained by Ribotta and Le Bailc [32].

**Table 5.** Freezable water in investigated wheat and wheat-kale (KP) composite bread.

| Sample | Day | $T_{on}$ (°C) | $T_p$ (°C) | $T_{end}$ (°C) | $-\Delta H$ (J·g$^{-1}$) | FW (g·g$^{-1}$) |
|---|---|---|---|---|---|---|
| Control | 1 | $-9.0 \pm 0.1$ [a] | $3.2 \pm 0.1$ [c] | $10.3 \pm 0.1$ [bc] | $76.7 \pm 1.7$ [d] | $0.230 \pm 0.005$ [d] |
| | 2 | $-9.1 \pm 0.1$ [a] | $2.5 \pm 0.7$ [bc] | $9.6 \pm 1.3$ [bc] | $65.5 \pm 3.4$ [c] | $0.196 \pm 0.010$ [c] |
| | 3 | $-9.1 \pm 0.1$ [a] | $1.1 \pm 0.4$ [a] | $7.3 \pm 0.0$ [a] | $49.8 \pm 8.1$ [a] | $0.149 \pm 0.024$ [a] |
| 5% KP | 1 | $-9.5 \pm 0.2$ [a] | $3.5 \pm 0.4$ [c] | $11.2 \pm 0.7$ [c] | $84.9 \pm 4.0$ [de] | $0.255 \pm 0.012$ [de] |
| | 2 | $-9.9 \pm 0.3$ [a] | $1.7 \pm 0.4$ [ab] | $8.9 \pm 0.8$ [ab] | $60.6 \pm 0.9$ [bc] | $0.182 \pm 0.003$ [bc] |
| | 3 | $-9.8 \pm 0.2$ [a] | $1.2 \pm 0.6$ [a] | $7.5 \pm 0.8$ [a] | $52.1 \pm 1.9$ [ab] | $0.156 \pm 0.006$ [ab] |
| 10% KP | 1 | $-9.3 \pm 0.9$ [a] | $3.7 \pm 0.6$ [c] | $11.4 \pm 1.1$ [c] | $88.6 \pm 8.3$ [e] | $0.266 \pm 0.025$ [e] |
| | 2 | $-9.6 \pm 0.6$ [a] | $3.4 \pm 0.7$ [c] | $10.9 \pm 0.8$ [c] | $87.7 \pm 3.4$ [e] | $0.263 \pm 0.010$ [e] |
| | 3 | $-10.3 \pm 0.1$ [a] | $2.8 \pm 0.2$ [bc] | $10.1 \pm 0.3$ [bc] | $76.3 \pm 1.6$ [d] | $0.229 \pm 0.005$ [d] |
| Two-way ANOVA-*p* | | | | | | |
| Factor A (level) | | 0.023 | 0.005 | 0.007 | <0.001 | <0.001 |
| Factor B (time) | | 0.172 | 0.001 | 0.001 | <0.001 | <0.001 |
| Factor A × Factor B | | 0.403 | 0.091 | 0.110 | 0.033 | 0.033 |

Mean value of two replication ± standard deviation. Means in columns marked with the same letters do not differ statistically at a 0.05 level of confidence.

During the heating of the samples, the melting of recrystallized amylopectin occurs connected with endothermic transformation, the presence of which is associated with the retrogradation of starch polymers [49]. The onset temperature of this transformation ranged from 50.7–54.1 °C, while the temperature values at the maximum ranged from 63.1 to 65.2 °C, with no significant variation between the samples studied. In turn, the values of the final transformation temperature ranged from 70.0 to 77.6 °C, with only the time of storage having a significant ($p < 0.05$) effect on its values. The determined temperature ranges confirm the previous literature data [48,49]. The two-factor analysis of variance performed showed that only time of storage had a significant ($p < 0.05$) effect on the determined values. On the first day, the sample with the largest proportion of lyophilized kale had the highest enthalpy value of retrograded amylopectin distribution, but the differences between samples were non-significant ($p > 0.05$) (Figure 5).

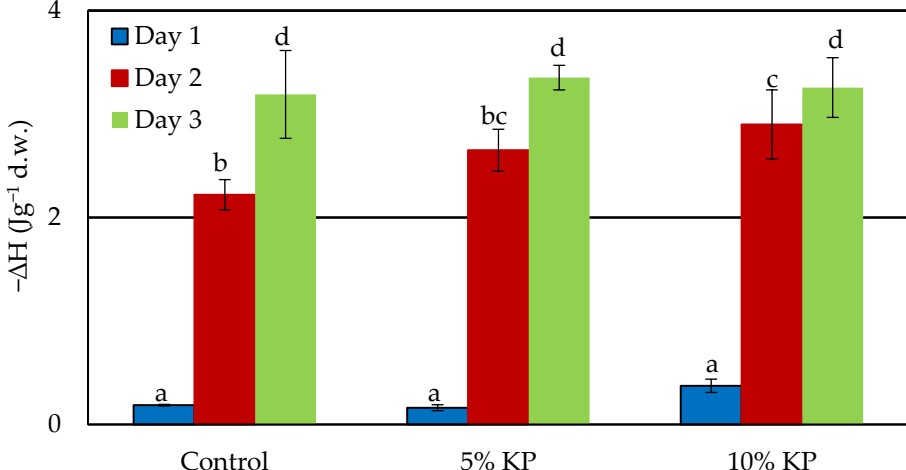

**Figure 5.** Effect of storage time on the enthalpy of amylopectin melting after retrogradation of the investigated wheat and wheat-kale (KP) composite bread; means marked with the same letters do not differ statistically at 0.05 level of confidence.

Similarly, on the second day, the crumb of bread enriched at the highest level showed the highest enthalpy, the value of which was significantly higher than that of the control. On the third day, no significant differences in the enthalpy values of the distribution of retrograded amylopectin were noted. The results obtained do not fully correlate with the analysis of crumb texture, which clearly indicates the progressive aging of the bread. The addition of lyophilized kale may modify this process while remaining unaffected by the retrogradation of amylopectin determined during thermal analysis. At the same time, the progressive hardening of the crumb is probably more related to the retrogradation of amylose or its complexes with high-molecular-weight components, as well as the loss of freezable water.

## 4. Conclusions

The results obtained clearly indicate the dependence of the observed rheological properties on the magnitude of the applied stress. An increase in the values of storage and loss moduli was observed in the range of small deformations with a decrease in the value of the phase shift angle, as well as a decrease in the value of compliance with an increase in zero shear viscosity value. The observed strengthening of the dough structure is due to the content in the system of high-molecular-weight components with high swelling capacity and competitive water absorption. On the other hand, an increase in the mixing tolerance index and dough development time was noted in the high strain range, indicating a deterioration of the dough strength. The weakened dough structure is due to less gluten in the system and limited hydration of gluten proteins. The addition of freeze-dried kale significantly increased the content of dietary fiber in the tested bread, both the soluble and insoluble fractions. However, the content of bioactive compounds increased the most-carotenoids and polyphenols. The addition of lyophilized kale caused a reduction in pore size and the number of large pores in the crumb while increasing the pore density. The changes in crumb structure characteristics appear to be favorable, but bread enriched with lyophilized kale was characterized by poorer acceptability of sensory attributes due to a significantly darker color with a distinctly greenish hue caused by the presence of chlorophylls. The share of kale powder in the recipe caused adverse changes in crumb texture. A significant increase in hardness was found, but a decrease in springiness, cohesiveness, and resilience. Introducing powdered vegetables into bread seems to be a good strategy for improving its nutritional value, but changes in quality result in a decrease in sensory acceptance. Therefore, proper health information is needed to promote such products.

**Author Contributions:** Conceptualization, A.K.; methodology, A.K., M.W., J.K. and L.J.; investigation, J.K., A.K., M.W. and L.J.; writing—original draft preparation, L.J.; writing—review and editing, J.K., M.W. and A.K.; supervision, J.K. All authors have read and agreed to the published version of the manuscript.

**Funding:** This research received no external funding. The work was financed by a subsidy of the Ministry of Education and Science Republic of Poland for the University of Agriculture in Krakow.

**Institutional Review Board Statement:** Not applicable.

**Informed Consent Statement:** Not applicable.

**Data Availability Statement:** Not applicable.

**Conflicts of Interest:** The authors declare no conflict of interest.

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
