# Peer review of "Dough Rheological Properties and Characteristics of Wheat Bread with the Addition of Lyophilized Kale (Brassica oleracea L. var. sabellica) Powder"

_applsci, doi:10.3390/app13010029_

Round 1
Reviewer 1 Report
The authors have done a fair job in designing the manuscript. The authors should address following queries:
1. Introduction: The authors introduce that there is a need to fortify protein and fibres through plant based products in the bakery foods, howeever, states kale enrich thee antioxidants and minerals and never addresses the protein and fibre issue.
Methodology: No reference to proximate/nutritional composition of the product made.
Results: Nutritional composition should be done and discussed in reference to other properties.
Conclusion: improve accordingly
references: adequate
Author Response
The authors thank the reviewer for valuable comments that improved the quality of the manuscript. Responses to individual comments are provided below.
The authors have done a fair job in designing the manuscript. The authors should address following queries:
1. Introduction: The authors introduce that there is a need to fortify protein and fibres through plant based products in the bakery foods, howeever, states kale enrich thee antioxidants and minerals and never addresses the protein and fibre issue.
The text has been supplemented accordingly.
2. Methodology: No reference to proximate/nutritional composition of the product made.
The reference has been added.
3. Results: Nutritional composition should be done and discussed in reference to other properties.
The text has been supplemented accordingly.
4. Conclusion: improve accordingly
The text has been supplemented accordingly.
references: adequate
Reviewer 2 Report
Comments and Suggestions for Authors
The manuscript is great. But, there are some comments
1-The calculated freezeable water content must be added to the materials
and methods.
2- Change Percentage of pores >5 mm2 to Percentage of pores >5 mm
3- Added Table 4 in the text in Bread texture
4- The statistical analysis should be added to the Table3.

Author Response
The authors thank the reviewer for valuable comments that improved the quality of the manuscript. Responses to individual comments are provided below.
Comments and Suggestions for Authors
The manuscript is great. But, there are some comments
1-The calculated freezeable water content must be added to the materials
and methods.
Freezable water calculation method has been added.
2- Change Percentage of pores >5 mm2 to Percentage of pores >5 mm
In the image analysis, the pore area was analyzed, therefore both the average pore size and pores >5mm2 should be expressed in a unit area. The text and Table 2 have been corrected accordingly, and "diameter" has been deleted from the section 2.4 in Methods.
3- Added Table 4 in the text in Bread texture
The table was cited at the end of the chapter, the citation was added also at the beginning of the chapter.
4- The statistical analysis should be added to the Table3.
Statistical analysis has been added.